# Decomposing a San Francisco estuary microbiome using long-read metagenomics reveals species- and strain-level dominance from picoeukaryotes to viruses

Lauren M. Lui,[1] Torben N. Nielsen[1]

**ABSTRACT** Although long-read sequencing has enabled obtaining high-quality and complete genomes from metagenomes, many challenges still remain to completely decompose a metagenome into its constituent prokaryotic and viral genomes. This study focuses on decomposing an estuarine metagenome to obtain a more accurate estimate of microbial diversity. To achieve this, we developed a new bead-based DNA extraction method, a novel bin refinement method, and obtained 150 Gbp of Nanopore sequencing. We estimate that there are ~500 bacterial and archaeal species in our sample and obtained 68 high-quality bins (>90% complete, <5% contamination, ≤5 contigs, contig length of >100 kbp, and all ribosomal and tRNA genes). We also obtained many contigs of picoeukaryotes, environmental DNA of larger eukaryotes such as mammals, and complete mitochondrial and chloroplast genomes and detected ~40,000 viral populations. Our analysis indicates that there are only a few strains that comprise most of the species abundances.

**IMPORTANCE** Ocean and estuarine microbiomes play critical roles in global element cycling and ecosystem function. Despite the importance of these microbial communities, many species still have not been cultured in the lab. Environmental sequencing is the primary way the function and population dynamics of these communities can be studied. Long-read sequencing provides an avenue to overcome limitations of short-read technologies to obtain complete microbial genomes but comes with its own technical challenges, such as needed sequencing depth and obtaining high-quality DNA. We present here new sampling and bioinformatics methods to attempt decomposing an estuarine microbiome into its constituent genomes. Our results suggest there are only a few strains that comprise most of the species abundances from viruses to picoeukaryotes, and to fully decompose a metagenome of this diversity requires 1 Tbp of long-read sequencing. We anticipate that as long-read sequencing technologies continue to improve, less sequencing will be needed.

**KEYWORDS** metagenomics, long-read sequencing, estuarine microbiomes, eDNA, genome binning, Nanopore, viruses

The effect of climate change on microorganisms and their potential to positively or negatively affect it is of growing importance (1–3). Ocean and estuarine microbiomes are of major interest because of their contributions to greenhouse gas production and consumption, as well as their fundamental roles in global element cycling (1, 2). For example, even though the ratio of the total biomass of marine phytoplankton to that of terrestrial flora is approximately 1:100, marine microbial communities contribute approximately half of net primary production (e.g., photosynthesis) of the Earth (4, 5). Despite the importance of modeling microbial communities to predict the effects of

Address correspondence to Lauren M. Lui, lmlui@lbl.gov, or Torben N. Nielsen, torben@jorgmundir.life.

Lauren M. Lui and Torben N. Nielsen contributed equally to this article. Author order is based on alphabetical order of last name.

The authors declare no conflict of interest.

See the funding table on p. 16.

climate change, a fundamental unit for helping us study environmental microbiomes still lies tantalizingly out of reach: complete genomes from uncultured microbes.

In the last 5 years, long-read sequencing (as developed by Oxford Nanopore Technologies [ONT] and Pacific Biosciences) has facilitated the assembly of complete or nearly complete genomes from microbial metagenomes from a variety of environments (6–14), as well as many plasmids and viral genomes (15–18). However, the data come with many technical and bioinformatics challenges. While it is known that high-quality, high-molecular-weight (HMW) DNA is needed for the best results from long-read sequencing (10), many groups use standard DNA extraction methods, which often result in suboptimal read length and quality. For environmental microbiomes, relic DNA is also an issue because not only does it represent dead organisms, the DNA fragments tend to be short (<500 bp) (19). Although longer reads can help overcome genomic repeats that fragment assemblies (20, 21), binning algorithms are generally not designed to handle the longer contigs and may fail to bin properly due to their reliance on accurate coverage measures (3, 22). Effectively using long-read sequencing for environmental metagenomics still needs to overcome the challenges of obtaining enough high-quality high-molecular weight DNA and improving bioinformatics methods.

In this study, we sought to establish methods to obtain complete or high-quality microbial and viral genomes using long-read metagenomics to study the microbiome of the San Francisco Estuary (SFE), the largest estuary on the West Coast of the United States. The SFE has high nutrient loadings, especially nitrogen and phosphorus, that are higher than other estuaries already impaired by eutrophication syndrome (23). This, along with increased algal toxins and primary production in recent years, supports the hypothesis that the SFE is at a critical tipping point, and better models are urgently needed to investigate the health of the ecosystem. Despite the importance of microbial populations in many global element cycles, including nitrogen and phosphorus, biogeochemical models of the estuary only coarsely represent microbial inputs and outputs (23) or only focus on phytoplankton (24, 25), and all ignore the viral populations. This may partially be due to the surprisingly few metagenomics studies of the SFE (3, 26–30). There appear to be fewer than 15 shotgun metagenomics studies, and most are focused on wetlands or microcystis blooms. More metagenomics studies could provide more information about which organisms are involved in element transformations that could be incorporated into models. We seek not only to fill this data need for the SFE but also to optimize long-read metagenomics methods to provide high-quality, complete microbial genomes for biogeochemical cycling studies. In this pilot study, we also specifically developed methods to study the viral and microbial communities at the same time, given the importance of viruses in marine microbiomes (31, 32).

Here we present a first look at deep Nanopore sequencing of a sample from the South San Francisco Bay (Fig. 1A). We sequenced this sample with 150 Gbp of Nanopore sequencing, which is a deeper long-read sequencing than any other marine or estuarine study that we know of and deeper than most long-read microbiome studies. We developed our own filtering and HMW DNA extraction protocols to improve the quality and amount of DNA available for sequencing (Fig. 1B), especially to have enough for size selection. Using a novel binning method (Fig. 1C), we were able to complete or nearly complete 68 microbial genomes, as well as identify ~1,900 plasmids and ~40,000 viral populations. We define near complete microbial genomes as greater than 90% complete, less than 5% contamination, no more than five contigs, contigs with a length of >100 kbp, and having full ribosomal gene complements and a full complement of tRNAs. This definition extends the high-quality draft definition previously published (33). In addition to the bacterial and archaeal members, we also generated partial genomes for several eukaryotes and obtained multiple full-length mitochondrial and chloroplast genomes.

We aimed to demonstrate what is possible with long-read metagenomics with better DNA extraction, sequencing depth, and improved bioinformatics methods. Specifically, we sought to ask for our sample (i) can we taxonomically classify all the contigs, (ii) can we accurately estimate the number of bacterial and archaeal genomes, and (iii) can we

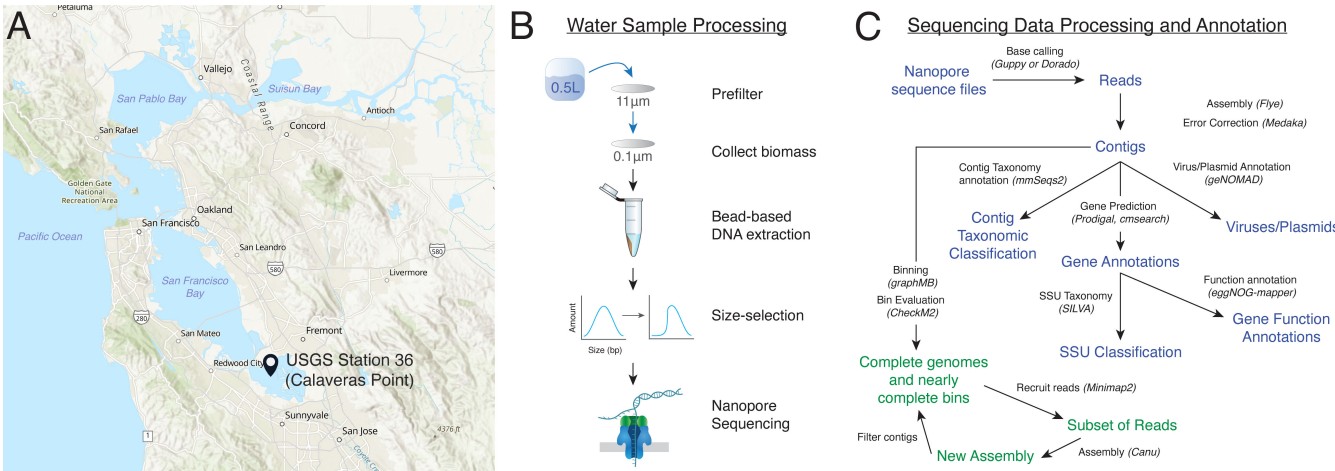

**FIG 1** Overview of sample and data processing. (A) Map of the San Francisco Estuary and the location of United States Geological Survey (USGS) station 36. Background map created using ArcGIS software by Esri with the baseline map "World Street Map (with Relief) - no labels" (ID: 5514df296eb348aeac7bf0006663cf48). (B) Overview of water sample processing and DNA extraction. (C) Overview of data processing of Nanopore sequencing data (blue) and iterative binning method (green). SSU, small subunit.

fully assemble all of the bacterial and archaeal genomes? Although we were not able to complete all of the genomes in our sample, we gained valuable information and determined that we have likely obtained most of the diversity in the sample and genome bins for most prokaryotic species present, approximately 40% of our sample consists of viruses, and we can estimate the depth of sequencing needed to separate strains and complete all of the genomes in our samples. These results represent advances in using long reads to obtain microbial genomes from metagenomes and an initial look into our project to decompose the entire SFE microbiome.

## RESULTS AND DISCUSSION

### Processing, sequencing, and assembly of water samples from USGS Station 36

We received water samples from Station 36 in the South San Francisco Bay (Fig. 1A). Station 36 is one of multiple USGS monitoring stations in San Francisco Bay, and the sampling methods are described in reference 34. Salinity at the station was 32.70 practical salinity units, and the temperature was 21.47°C.

To improve collection of the entire microbial population (<11 µm), we chose to collect biomass directly onto a 0.1-µm filter after prefiltering (Fig. 1B), rather than sequential filtering as is done in many other water metagenomics studies (35). This reduces possible loss from sequential filtering and allows us to capture ultra-small bacteria and viruses (36). To extract high-quality HMW DNA for long-read sequencing, we developed a method that uses gentle enzymatic lysis and carboxyl beads to reduce DNA shearing and maximize DNA quality ($OD_{600}$ 260:230 and 260:280 >1.8).

We initially sequenced Nanopore libraries on two R9.4.1 flow cells but found that we wanted deeper sequencing to improve coverage of some low-abundance species and thus improve their assemblies. We obtained additional sequencing with two R10.4 PromethION flow cells for a total of ~150 Gbp of data. We obtained a read N50 of 11,935 bp, which is about twice that of other long-read environmental water metagenomes (N50s between 1,000 and 6,000 bp) (8, 14, 37). This Nanopore sequencing data were basecalled using Guppy (Oxford Nanopore Technologies, UK), assembled using Flye (38), and polished with Medaka (Oxford Nanopore Technologies). We also obtained 20 Gbp of Illumina sequencing from the same DNA used for the Nanopore libraries. This Illumina sequencing data were quality filtered and trimmed before being used as input to SPAdes (39).

## Taxonomic composition of the sample at the superkingdom level

Theoretically, the longer contigs afforded by Nanopore sequencing provide more sequence and gene resolution on which to base classification, rather than relying on marker genes. Although the focus of this study was to assemble high-quality bacterial and archaeal genomes, we also attempted to classify all 107,977 contigs in the sample at least to the superkingdom (domain) level. This effort served to provide an overview of the organisms that we captured on our filters (0.1–11.0 µm) across all three domains of life and to demonstrate how well we could classify contigs that were not bacterial and archaeal, which interferes with relative abundance estimates (40).

We used three classification methods in conjunction with manual curation to analyze the contigs in our samples: (i) geNomad, a deep neural network method, to classify contigs as mobile genetic elements (i.e., plasmids and viruses) and to assign viral taxonomy; (ii) MMseqs2, which assigns taxonomy based on protein genes, with Genome Taxonomy Database (GTDB) as the reference database (so classification was biased toward bacteria and archaea); and (iii) Kaiju to classify eukaryotic contigs using the Kaiju database that contains a subset of the National Center for Biotechnology Information (NCBI) BLAST nr database containing all proteins from archaea, bacteria, viruses, fungal, and microeukaryotic reference genomes (db_nr_euk). Manual curation was used to detect spuriously assigned contigs based on the small subunit ribosomal ribonucleic acid genes (small subunit [SSU] rRNAs). For example, a contig could be classified as viral but has a bacterial SSU. In these cases, we classified contigs as eukaryotic, mitochondrial, or chloroplast by using NCBI BLAST (41) on the SSU genes.

Not unexpectedly, using multiple classification methods leads to discrepancies, but we found that the databases had the largest impact on the results (Supplemental Methods). If an organism's genome is not close to anything in the database, many of these methods attempt to determine the last common ancestor. However, often classifications at the superkingdom level (e.g., a sequence classified as bacteria with no phylum or lower taxonomic rank assigned) are ambiguous and get classified differently, depending on the algorithm and database. For example, if only MMseqs2 with the GTDB database was used, 85,455 contigs are classified as bacterial. However, if this is combined with the classification results for viruses and eukaryotic sequences, the number of bacterial contigs is halved to 40,986 contigs (Supplemental Methods). This underscores the importance of continuing to get complete genomes from metagenomes to better populate reference databases and to treat low confidence classifications with skepticism.

Combining all of the results led to the conclusion that of the contigs we assembled, 4% are eukaryotic; 0.3%–0.7% are archaeal; 38% are bacterial; 40% are viral; and 14% are unclassified/ambiguous (Fig. 2A). A more conservative estimate of the number of viral contigs is 25%, which also increases the number of unclassified contigs to 29%. We encourage the reader to see the Supplemental Methods for full results from each of the classification methods, assumptions, and reasoning for reaching our final classification of the contigs. The unclassified/ambiguous contigs had a median length of ~5,000 bp; possibly many of these contigs could not be classified because they were too short, rather than the databases not having representative sequences. Sorting through the different classification methods also points out the hazards of relying on one method for classification, even if the focus of a study is purely prokaryotic, viral, or eukaryotic. Where different methods agree, there is higher confidence that the classification is correct.

## *Bacterial taxa comprise nearly all prokaryotic contigs, while archaeal taxa are rare*

Unsurprisingly, given that we prefiltered the water at 11 µm, archaea and bacteria comprised a large proportion of the contigs. There were 40,986 bacterial contigs, which is 38% of all contigs. We only obtained one contig with a complete archaeal SSU gene, which was classified as in the *Nitrosopumilus* genus by both MMSeqs2 and SILVA (Tables S1 and S2). This is consistent with a recent 16S study where a *Nitrosopumilus*-like operational taxonomic unit (OTU) was the only dominant archaeal OTU in the South Bay

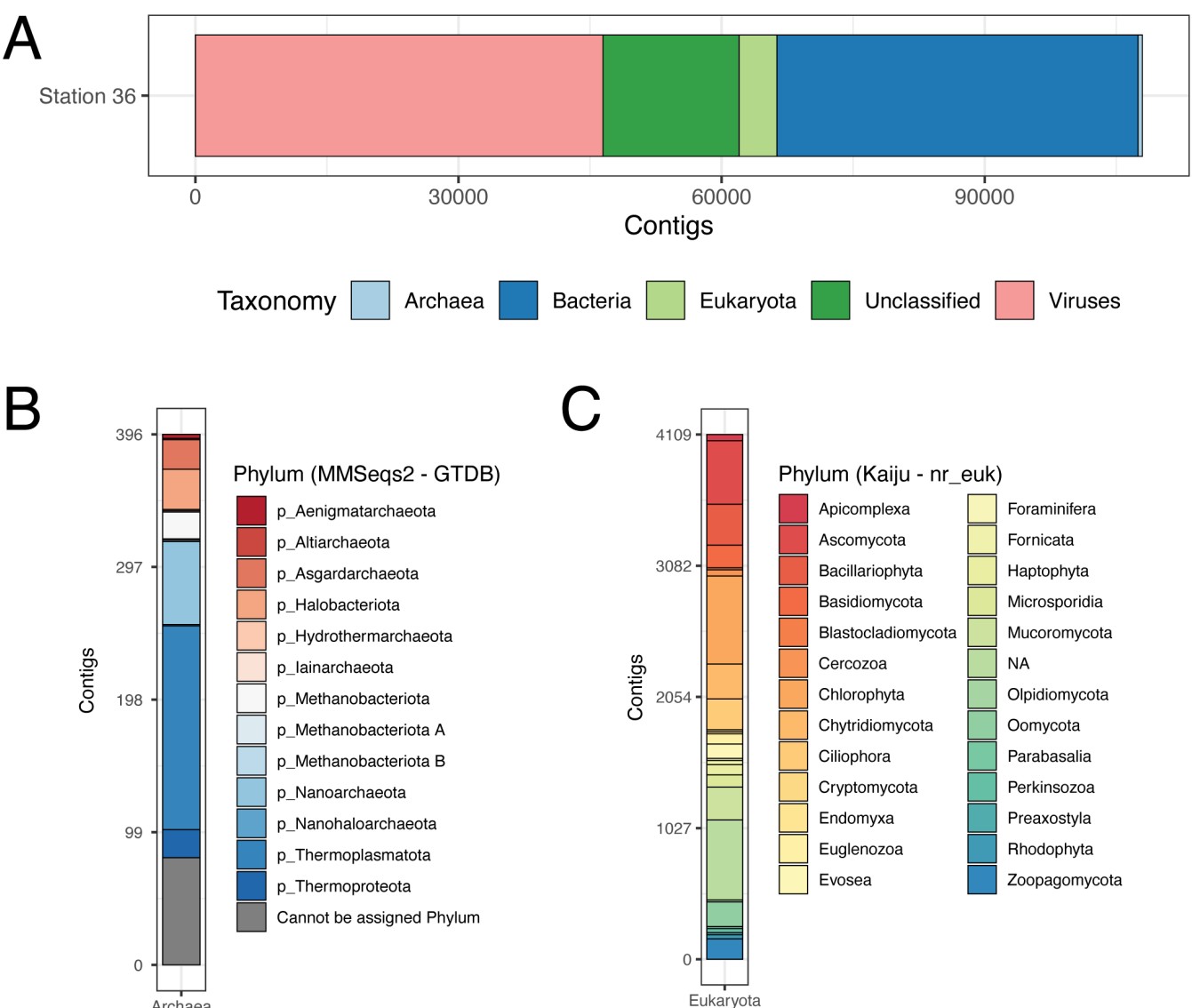

FIG 2 Contig classification. (A) Curated taxonomy at the superkingdom level based on geNomad, Kaiju, and MMseqs2 (Supplemental Methods). (B) Archaeal contig classification at the phylum level based on MMseqs2. Although there are only 14 *Nitrosopumilus* contigs (14 of the 21 Thermoproteota contigs in panel B), the mean length of the contigs is twice that of the other archaeal contigs (24,632 vs 12,916 bp). The longer contigs are consistent with the fact that the only full-length archaeal SSU is from *Nitrosopumilus*, and thus, the assembly of this genome is more contiguous than the other archaeal genomes present. (C) Eukaryotic contig classification at the phylum level based on Kaiju using the nr_euk database. Note that these results include only microeukaryotes and fungi (limitation due to Kaiju database). The eukaryotic contigs classified based on SSU analysis are not shown here because a different classification database was used (SILVA or manual curation based on NCBI nr database search).

of the SFE (42). However, the archaeal SSU gene in our data set likely represents a different species from that represented by the OTU or the two high-quality MAGs that Rasmussen et al. reported in a subsequent metagenomics study (26), based on an SSU alignment (Fig. S3). This suggests that there is still an unexplored location- or condition-dependent dominance of specific *Nitrosopumilus* strains in the South Bay.

Despite the existence of only one full-length archaeal SSU gene, MMseqs2 classified contigs in 12 other archaeal phyla (Fig. 2B). We are not prepared to assert that the archaeal diversity is as high as this. We used the GTDB taxonomy, which is sometimes based on highly fragmented and incomplete bins, and we considered some taxonomic collapse possible, i.e., the high number of closely related species in the GTDB database may have been caused by sequencing error and assembly artifacts. However, coverage

of the contigs involved is in the 5–10× range and thousands of bases in length, and we believe there is a significant archaeal population at USGS Station 36. We consider it likely that much of the population is in the sediment rather than being planktonic.

## Plasmids have a 3:1 ratio with archaeal and bacterial species

Based on the geNomad assignments, and removing the contigs that have SSUs, there are 1,963 plasmids. Since we estimate that there are 450–500 archaeal and bacterial species (see next section for justification of this estimate), plasmids appear to be at a 3:1 ratio with the number of genomes. This ratio underscores the importance of ongoing efforts to link plasmids to chromosomes in metagenomes (15, 43). Of the plasmids that had MMseqs2 classification phylum level or below, three of the contigs were classified at the archaeal and 1,591 were classified as bacterial, consistent with the estimated ratio of archaeal to bacterial species.

Plasmids are thought to be better retained by Illumina sequencing due to size selection and the inability to sequence circular contigs with Nanopore native ligation library preparation. However, in the Illumina assembly, geNomad identified only 244 contigs (average length, 3,663 bp) as plasmids, as compared to the 1,963 plasmids it identified in the Nanopore assembly (average length 15,264 bp) (Tables S3 and S4). Since the Illumina sequencing was about 14% of the depth of the Nanopore assembly (21 Gbp vs 150 Gbp) and the assembly is about a third of the Nanopore assembly (0.95 Gbp vs 2.95 Gbp), fewer identified plasmids are expected in the Illumina assembly. However, in terms of the number of total contigs in the assemblies (813,394 vs 107,977 contigs of Illumina vs Nanopore assemblies), the fraction of contigs identified as plasmids is far fewer in the Illumina assembly (0.03% vs 1.8%). In addition, the shorter length of contigs likely led to generally lower confidence predictions by geNomad (Fig. S4). From this analysis, despite using both size selection and native ligation in the Nanopore library preparation, we obtained more plasmids in the Nanopore assembly, and the Nanopore assembly allows for higher confidence identification of plasmids. This result may be due to the extremely deep Nanopore sequencing, although 21 Gbp for an Illumina metagenome is generally deeper than what other studies obtain.

## Eukaryotic species, mitochondria, and chloroplasts

Using the eukaryotic specific database with the Kaiju classification of contigs, as well as manual curation of eukaryotic, mitochondrial, and chloroplast SSUs, allowed us to determine that we sequenced environmental DNA of eukaryotes, such as mammals, polychaete worms, clams, mussels, sea anemone, zooplankton, ciliates, and diatoms (Table S1; Fig. S5). We also detected smaller eukaryotes that came through the 11-µm prefilter, such as *Ostreococcus tauri*. The top five species based on contig counts classified by Kaiju were ciliate *Stylonychia lemnae* (896 contigs), diatom *Thalassiosira pseudonana* (524 contigs), green alga *Ostreococcus* sp. "*Lucimarinus*" (396 contigs), green alga *Picochlorum* sp. BPE23 (273 contigs), and *Ostreococcus tauri* (272 contigs). Eleven species out of 256 comprised 82% of the eukaryotic contigs classified by Kaiju (Table S5). The Kaiju database only had eukaryotic sequences from fungi and microeukaryotes and did not report classifications from larger organisms.

We assembled some complete, circular mitochondria and chloroplast genomes and classified them based on their SSUs (Table S1). In general, nearly all chloroplasts were initially classified as cyanobacteria (19 of 22) based on SILVA. Only one mitochondrion was correctly classified by SILVA; the rest needed manual curation. We found that if we had only relied on the MMseqs2 classification of contigs, we would have misclassified the mitochondria and chloroplasts as bacteria. Most were classified as bacteria at the superkingdom level, but four of the mitochondria were classified at the species level (UBA4416 sp016787365, PWPS01 sp003554915, and CAIVPM01 sp018335935) and two at the genus level (*Pelagibacter* and UBA2645). Similarly, there are some chloroplasts that are classified in the class Cyanobacteria. Current theory holds that mitochondria arose from a bacterium becoming an endosymbiont in an archaeal cell (44), so similarities of

between these genomes and bacterial ones are not unexpected. However, these results are a reminder to check if a MAG from a metagenome is a mitochondrion or a chloroplast before labeling it as a new species.

## Estimation of the number of prokaryotic genomes at USGS Station 36

Key questions in metagenomics concern what organisms are present in a sample and what the relative taxonomic abundances are. Not only do the answers to these questions provide an idea of the types of functions occurring in the sample, but also they provide species abundances to analyze with environmental data. In order to estimate how many genomes we could aim to assemble, we used marker genes to estimate the number of prokaryotic genomes present and then we used the coverage of the contigs as proxies for relative abundance.

Two common choices for marker genes are the SSU ribosomal RNA gene and 30S ribosomal protein S3 (rpsC) (45). SSUs have the advantage that they are the basis for much of modern taxonomy (46) prior to the advent of GTDB (47) using full genomes or genome bins for taxonomic classification, and there are existing databases that can be used to place SSUs in the Tree of Life (48, 49). At the same time, the disadvantage to using SSUs is that sometimes they have copy numbers greater than one, which distorts relative abundance measurements. Using rpsC can overcome this issue since it is a single-copy gene, but the lack of databases providing taxonomic information is problematic. Nanopore sequencing can mitigate these issues because the reads are long enough to span ribosomal operons; thus, SSUs (and the rest of the ribosomal operon) do not cause assembly fragmentation (50). Moreover, if the sequencing is deep enough, the contigs are often sufficiently long to contain both SSU and rpsC genes, thus allowing rpsC genes to be assigned a taxonomy consistent with that of the SSUs.

To evaluate how much of the microbiome we were able to decompose into individual genomes, we used SSUs, rpsC, and contig taxonomy via MMSeqs2 (using GTDB as the database) to get a count of distinct organisms. First, we evaluated the SSUs in the Nanopore assembly. We found a total of 561 bacterial and archaeal SSUs. Of those, 37 were reported as truncated by *cmsearch*; i.e., the covariance model could not be fit at one or both ends. We manually curated the 37 and concluded that they were assembly artifacts or from extremely rare organisms and decided to exclude them from further consideration. This left 524 SSUs.

There are two confounding factors when using SSUs to estimate genome abundances: (i) one genome can have multiple SSUs that have different sequences, and (ii) genomes from different species or strains can have identical SSUs (51). Accounting for contigs that contain multiple SSU sequences (29; Fig. 3; Table S2), we found that the number of estimated genomes becomes 495. We found 17 contigs that have at least two different SSU sequences. Searching for identical SSUs between different contigs, we found 49 groups of SSUs. We did not reduce the number of estimated genomes based on contigs having identical SSUs because it is not possible to tell if these contigs belong to the same genome or to different strains. For example, two *Glaciecola* contigs, contig_37285 (772,808 bp) and contig_57621 (575,645 bp), both have two SSUs, and all four of these SSUs are identical. However, an alignment of these two contigs using LAST (52) did not result in any significant alignment longer than 5,743 bp, which is likely the alignment of ribosomal operons. This suggests that these contigs either represent two pieces of the same genome or they are from different strains/species, despite having identical SSUs. For these reasons, some of the SSUs in Fig. 3 are identical, and our estimate of the number of genomes based on SSUs is still 495 but with a potential lower bound of 447.

We next annotated all of the predicted protein-coding genes in the Nanopore assembly and extracted the rpsC genes. We found a total of 548 contigs with rpsC genes. However, the Nanopore assembly also contained genomes for 59 mitochondria and 23 chloroplasts, and these generally contain rpsC genes as well. Eighteen of the rpsC genes

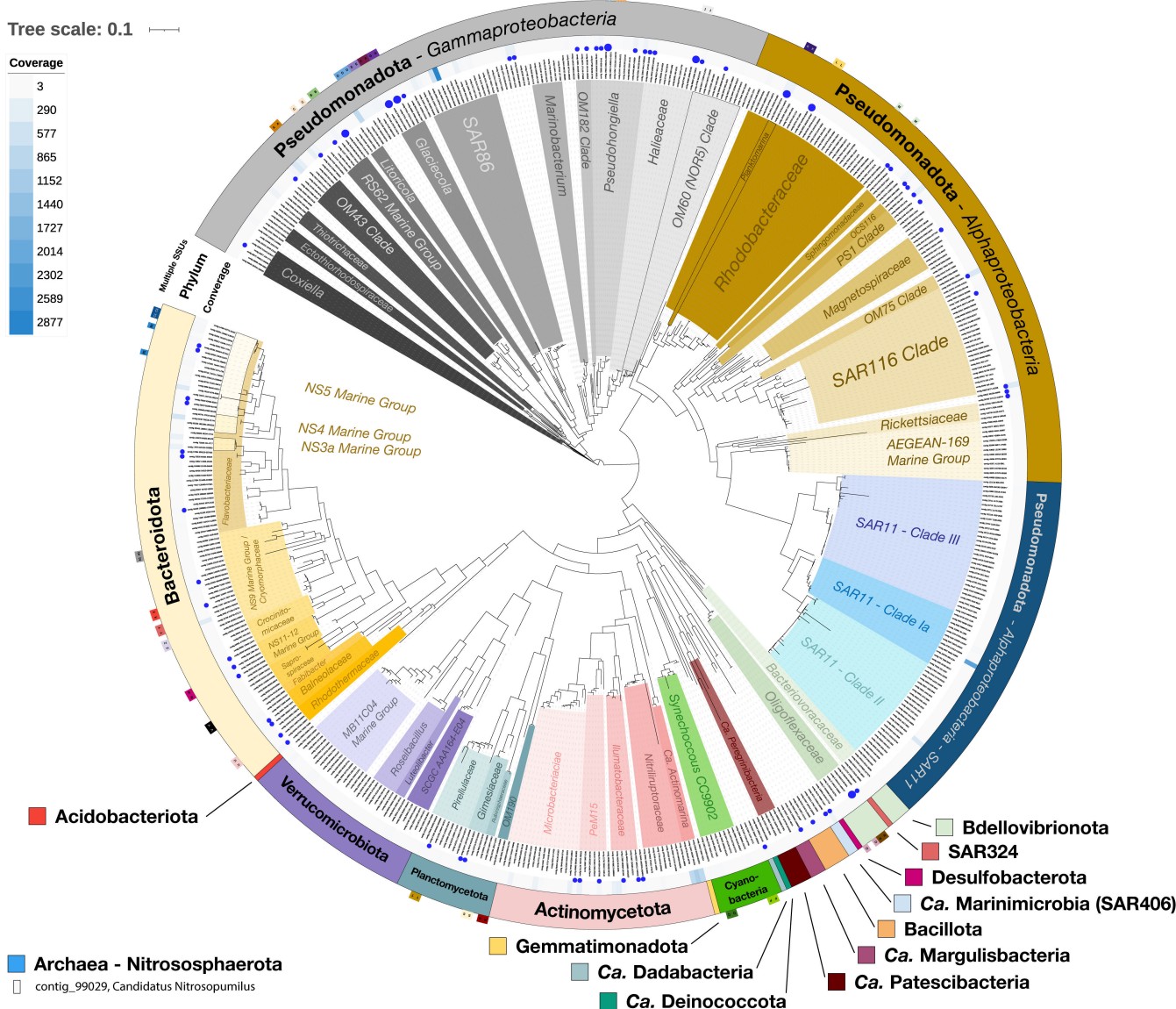

**FIG 3** Taxonomy for full-length bacterial and archaeal SSUs. A tree of the bacterial SSUs is shown. There was only one full-length archaeal SSU, which is shown in the bottom left corner. The outer ring of the tree indicates if there is a pair of SSUs on the same contig (see Table S2 for more details); the next ring indicates the phylum; and the third ring indicates the coverage of the contig the SSU is on. The blue circles indicate that a high-quality bin exists with that SSU; larger circles are used for contigs that have more than one SSU. The inner taxonomy was added manually to highlight family-, genus-, or species-level groups of interest. The coverage of the contig with the archaeal SSU is also indicated on the same scale as the bacterial tree (coverage of eight).

were on contigs known to have mitochondrial SSUs, so the upper bound on the number of prokaryotic genomes represented by rpsC genes is 530, and the lower bound is 466.

Finally, we used MMseqs2 taxonomy on all of our contigs with the GTDB database. We selected the contigs with species-level taxonomic assignments, at least 10 labels that agreed with the contig assignment, and coverage of at least 10 in order to get a conservative taxonomic assignment. This yielded an estimate of 489 as a lower bound for the number of distinct species. Note that this taxonomic assignment amounts to a clustering of the longer contigs using the GTDB organisms as seeds. A set of contigs all being assigned to the same GTDB organism at the species level does not necessarily mean that it is the same species or strain; it may just be a statement that the contigs are closer to the GTDB organism than to anything else in the database, and we infer that they are therefore likely to be close to each other as well.

Based on the three estimates, we conclude that the USGS Station 36 sample contained approximately 450–500 distinct bacterial organisms to which we can confidently assign taxonomy. Table S2 lists the organisms in decreasing order of relative abundance, i.e., coverage of the contig. We also examined diversity based on the Illumina assembly and found a total of 467 prokaryotic SSUs. However, only 36 were not found to be truncated by cmsearch. It is worth noting that while the estimate based on the Illumina assembly leads to a consistent estimate of the number of distinct species, it does not generally allow assignment to lower taxonomic ranks (Table S6). Most of the fragments were significantly shorter than 1,000 bp, and only the extreme sensitivity of cmsearch allowed them to be seen (53). A comparison of the taxonomy at the class level between the Illumina and Nanopore SSUs generally shows agreement for the number of prokaryotic SSUs that can be classified (Fig. S6), although ~42% of the total Illumina SSUs are unclassified, likely because they are truncated.

We generated a phylogenetic tree based on the bacterial and archaeal SSUs to study the phylogeny and included coverage of the contig the SSU was on as a proxy for abundance (Fig. 3). Six species comprised 30% of the relative abundance of bacteria and archaea: a SAR86 species (10.3%), a SAR11 Clade Ia species (7.8%), three *Actinomarina* species (3.9%, 3.0%, and 2.9%), and a *Litoricola* species (2.8%). Notably, all of these species except for the *Actinomarina* had closely related relatives that were not highly abundant. These results suggest that in order to study the population dynamics of the estuary microbiome, we will need to look at the individual genome level. Although we were able to tell which species were the most abundant, obtaining high-quality genome bins is necessary to help understand why some may be more successful than others.

## Binning of prokaryotic genomes and bin refinement

Ideally, we would like to extract the genome of each of the ~500 bacteria and archaea we believe to be present at Station 36 as one—generally circular—chromosome per contig. Six genomes were circularized immediately by the assembler, and we added a few by inspection. For the rest of the genomes, we needed to create bins. We were selective about choosing binning software since most are designed for Illumina-based assemblies and may not work well with Nanopore assemblies due to higher rates of sequencing error. Eventually, we settled on GraphMB as it employs graph neural networks that take full advantage of the assembly graph and long reads, which many other metagenome binners do not (54). To obtain high-quality genomes, we did initial binning with GraphMB, then did bin refinement by a novel method (Fig. 1C).

We used GraphMB with default parameters and then ranked the list of bins to find candidates for refinement. GraphMB produced a total of 1,310 bins. We used CheckM2 (55) analysis to help evaluate bin quality. Of the total, 128 bins met the requirement that completeness was greater than 90%. Of these, 59 bins also met the requirement that the contamination was less than 5%. All of the genomes reported to be circular by Flye were included in the 128 bins.

Next, we used a novel bin refinement method on these 128 bins. For short-read metagenomes, we previously published an algorithm called Jorg for improving bins to the point where they consisted of a single circular chromosome (21) and to detect misassemblies. The Jorg workflow first uses SPAdes for assembly and bins the contigs. Next, high-quality bins are picked for refinement. Reads that map to these bins are reassembled with MIRA, an overlap layout consensus (OLC) assembler, and often the contigs can be extended at this step, sometimes resulting in one complete, circular chromosome. The advantage of using SPAdes initially is that it is a k-mer-based assembler and is less memory intensive than MIRA. However, k-mer-based assemblers can cause misassemblies which can result in "chaos" bins. Reassembly of chaos bins with MIRA results in the contigs fragmenting into many pieces, indicating that the contigs were misassembled [see the Jorg manuscript for more details (21)].

We adapted the the Jorg workflow for long-read metagenomes (Fig. 1C). First, we used minimap2 (56) with default parameters to map the full set of long reads to the bin.

We then used the reads obtained as input to Canu (57) for reassembly. Canu is an OLC assembler and generally produces fewer misassemblies than Flye, but because of the computational requirements is too difficult to run on the entire set of Nanopore reads. We ran Canu with default parameters to produce a set of contigs which we manually curated based on GC, coverage, and CheckM2 results to obtain a new bin. We then iterated the process using the new bin.

The process was ended when there was no further clear improvement in either completeness, contamination, or number and length of contigs of the bin. For 68 bins, we were able to improve them to where they met our definition of nearly complete genomes, i.e., >90% complete, <5% contamination, no more than five contigs, a full ribosomal RNA gene complement, all necessary tRNAs, and no contigs shorter than 100 kbp (Table S7). Anecdotally, we would like to point out that of all 128 bins we worked on, only 1 failed due to the requirements of no more than five contigs and no contig shorter than 100 kbp, implying that in practice, these are not particularly onerous requirements.

This bin refinement method allows the merging of contigs and separation of highly similar strains. The example of contig_37285 and contig_57621 in the previous section, where both contigs have identical SSUs, demonstrated that these two contigs belong to the same genome. In other cases, we found that two genomes that have identical SSUs would separate, but whole genome alignments indicate that they should be separate strains (Fig. 4). The main determinant for success appears to be coverage. If coverage is adequate—at least ~20×—our bin improvement method works extremely well. The exception to that rule are organisms from the SAR11 clade, where widespread homologous recombination is believed to be the reason why it is difficult to obtain complete SAR11 genomes from metagenomes despite it being the most abundant marine bacterial group (58). SAR11 organisms are known to be very difficult to bin and assemble (58), and we expect to need significantly more sequencing and possibly isolation efforts to complete their genomes.

The methodology used for generating bins here is extremely flexible. Using it with contigs where coverage is ~40× or better, improvement is generally swift. A significant number of bins are high-quality drafts at the start and only need a reduction in the number of contigs and a restriction on the minimum length of contigs. A number of the nearly complete genomes we have are present as a single contig. We have also experimented with using single or a few long contigs we believed to belong together as starting material and then iterated to extend the contigs. In some cases, the process can be accelerated by looking for significant overlaps with other contigs in the assembly.

## Viruses compose ~42% of the Station 36 contigs

Marine viruses have a huge influence on the population dynamics of marine microbial populations, as well as element cycling in the marine environment (31). As mentioned

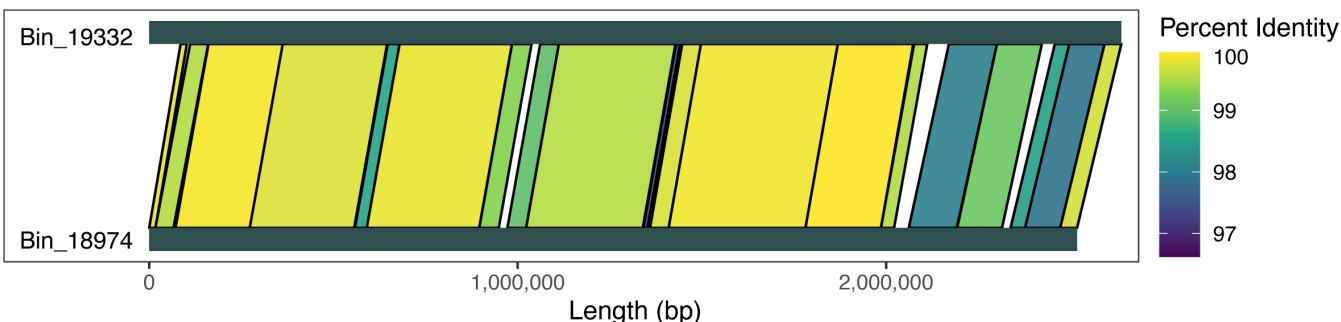

**FIG 4** Alignment of single-contig *Pseudohongiellaceae* bins with identical 16S sequences. Both bins are approximately 98% complete by CheckM2 analysis. This is an example of how our novel binning method can separate out closely related strains. Contigs were aligned with LAST (52), and the percent identity of the alignment regions is visualized. Although the contigs have identical 16S sequences, based on the alignment, they are closely related strains, and the differences are greater than you would expect from sequencing error.

in a previous section, viruses comprise a large part of the total contigs in our sample (Fig. 1A). We cannot account for RNA or single-stranded DNA viruses in this analysis, so the ratio of viruses to the other organisms is likely even higher. Using geNomad, we annotated 45,790 contigs as viruses (Fig. S7). Two of these had SSUs, so we reduced this number to 45,788. Given that the assembly had 107,977 contigs in total, viruses account for 42.4% of the contigs. Considering bases rather than contigs produces a very similar percentage. We did not expect such a high proportion of viruses, and we attribute it to using a 0.1-µm filter and letting it come very close to clogging.

To estimate the number of viral populations, we used the heuristic of ≥95% average nucleotide identity (ANI) as determined by an analysis of the Global Oceans Viromes 2.0 (GOV 2.0) data set (59). There were 18 sequences that were considered duplicates at ≥95% ANI and 611 sequences that were contained in other sequences, so we estimate the total number of viral populations to be 45,169 in our data set. Of these, 3,112 were found to be circular by the Flye assembler. The median length of the viral population sequences is 11,624 bp, and the mean is 23,243 bp (Fig. 5). Despite the modest average length of the viral population sequences, there are 88 longer than 300 kbp, 3 of which are greater than 1 Mbp, which puts them into the genome size of giant viruses (60). Of these, 20 are in the order *Caudoviricetes* and 14 are circular (Fig. S8). The rest are in the phylum *Bamfordvirae*, where all except one are classified in the order *Megaviricetes*, and eight are circular. The two longest circular viruses are 601,220 and 1,006,597 bp, and their lowest taxonomic classification is *Megaviricetes*.

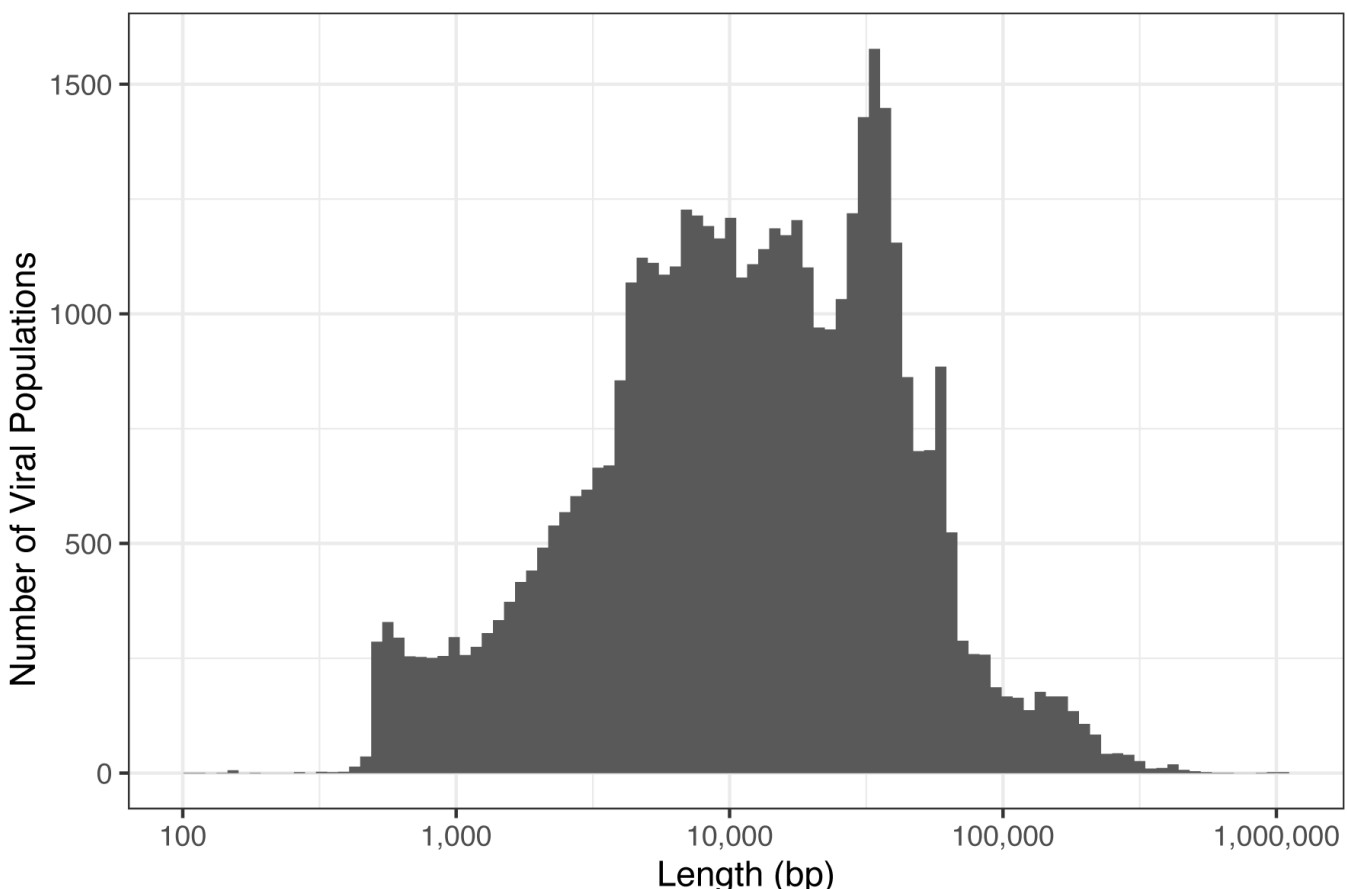

**FIG 5** Histogram of viral population contig length in the Station 36 metagenome. There are three viral realms represented in the data set (Fig. S6), although 5,281 could not be classified at the realm level and 2 populations were classified in the class *Naldaviricetes*, which does not have a higher taxonomic classification assigned to it. The realm *Duplodnaviria* comprised 80% of the viral populations, 99.9% of which was classified as *Caudoviricetes* (Fig. S6B). Only two viral populations were classified under the realm *Monodnaviria* (Fig. S6C). In the realm *Varidnaviria*, 3,787 populations were present in the sample, and 58% of this realm was classified under the family *Phycodnaviridae* (Fig. S6D).

The *Phycodnaviridae* contigs comprise approximately 4.8% of the total viral populations, and members of this family are the most abundant viral populations in this sample (Fig. 6). Members of this family are known to infect marine picoeukaryotic and microplankton species, such as *Ostreococcus*, and diatoms (60), so it is not unexpected that these would be highly abundant in our sample, and have been found to be abundant in other estuarine viromes (61). Twenty of the 2,195 *Phycodnaviridae* populations have coverage of >1,000 and range in length from ~31 to 236 kbp (Fig. S9). The dominance of 10% of the *Phycodnaviridae* populations in terms of abundance emphasizes the importance of determining the top abundant viral strains in a sample.

Determining the hosts of viruses is still difficult but is extremely important for modeling the marine microbiome population dynamics (31). We examined the intersection of the MMSeqs2 taxonomy with the geNomad predictions more closely, especially since 38,109 contigs of the 46,478 predicted to be viral by geNomad were classified as either archaea or bacteria by MMseqs2. We found that the top 5 longest of viral contigs classified as archaea by MMseqs2 were all circular and between 289,857 and 307,949 bp in length. Manual analysis of these contigs indicated that these were indeed archaea phage, as the presence of viral genes was spread through the entirety of the contigs. Specifically, the MMSeqs2 classification for all five is "d_Archaea; p_Nanoarchaeota; c_Nanoarchaeia; o_Pacearchaeales; f_GW2011-AR1," which suggests that these are a type of Nanoarchaeota phage. It is possible that we can use the MMSeqs2 classification to help determine hosts of some of these viruses, but this will require further analysis that is beyond the scope of this study. Plotting coverage of viruses by finer taxonomy indicates that there may be phage strain-specific dominance (Fig. S10), based on this hypothesis.

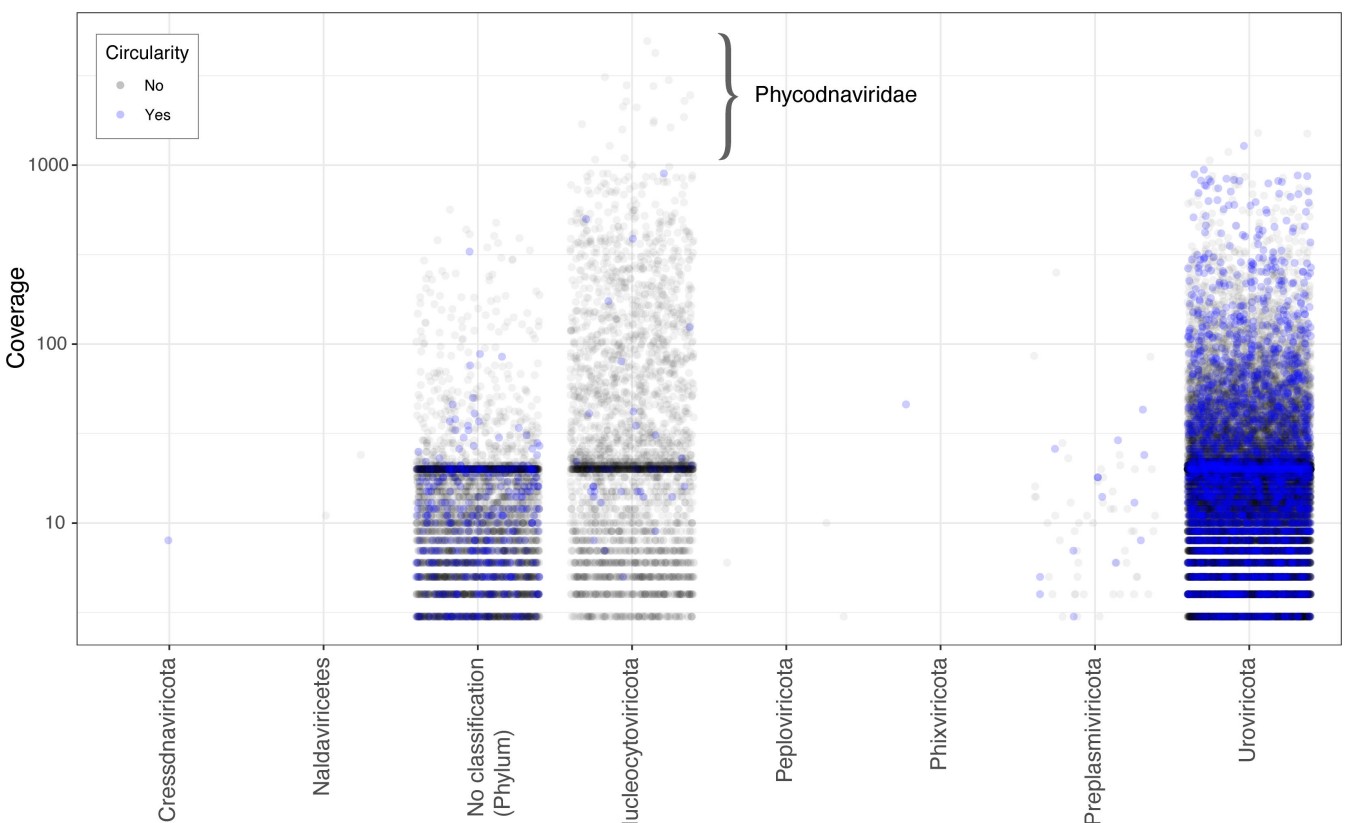

**FIG 6** Plot of coverage of the viral populations by phylum. Circular viral populations are in blue. In the *Nucleocytoviricota*, all of the viral populations with a coverage of >1,000 belong to the *Phycodnaviridae* family except one.

## Conclusion

In this study, we expended significant effort on determining exactly who is present down to the species/strain level, including eukaryotes and viruses, which was significantly improved compared to other studies due to our efforts to obtain high-quality sequencing data and the novel bin refinement method. In general, 20× coverage of a genome is considered necessary to ensure a high-quality assembly, at least based on analysis of Illumina metagenomes (62). Thus, to fully decompose this metagenome, we need to have at least six times more sequencing depth (3× was the lowest coverage of any full-length SSU), so approximately 1 Tbp. We would like to note that since we completed this study, Oxford Nanopore Technologies has come out with new basecalling models trained on prokaryotic data (December 2023), which we expect will result in reduced error rate and increase the contiguity of the assemblies. These results help indicate the sequencing depth other studies with similar samples may need if they want to obtain high-quality genomes from Nanopore metagenomes. Our focus on strain-level analysis helped enable a closer look at estimating the total number of prokaryotic genomes in the sample, but also a closer look at differences in abundance, including those for eukaryotic and viral species. The data from this study will be used for ongoing biogeochemical cycling studies of the SFE, but it also provides a resource to analyze historical 16S amplicon data to look at longer temporal population trends.

The biggest issue with our binning methodology is the amount of person hours it requires. We estimate that producing a nearly complete genome like this takes 5–10 person hours per genome. However, the quality of the final result is worth it to us. Our goal is to completely decompose the metagenomes in the SFE and to obtain complete genomes for as many of them as possible. As we move forward, we expect the work we do now will allow us to reduce the time investment needed. Obtaining complete genomes for all of the bacterial and archaeal species of the SFE will require a substantial sequencing effort. Our initial tranche is 5 Tbp of Nanopore long-read sequencing. We expect to (i) catalog the microbial diversity in the SFE; (ii) generate fully complete genomes of as many of the microbial species in the SFE as possible; and (iii) catalog mobile genetic elements in the SFE, such as viruses and plasmids. As this project unfolds, we expect to have enough data to produce a database that assigns taxonomy to rpsC genes based on the taxonomy of their SSUs. There have been multiple calls for long-term monitoring and study of marine microbiomes (1, 2), and this study is a first step to monitoring changes at the genome level.

## MATERIALS AND METHODS

### Water sampling and processing

Ten liters of surface water was collected by Erica Nejad and the crew of the USGS R/V David H. Peterson on 20 July 2022 from USGS Station 36 (Fig. 1A). We received the water on the same day and immediately transported it to a cold room kept at 4°C. Prefiltering through an 11-µm cellulose filter (Zenpore qualitative grade 1, ST001-125) to remove particulate matter and eukaryotes was done on the same day. Final filtering through 0.1-µm polyethersulfone (PES) filters (Pall, 60311) was carried out within 24 hours. We passed 0.5 L of water through each of 20 0.1-µm filters and immediately transferred them to a −20°C freezer.

### DNA extraction and size selection

We extracted 13.7 µg of HMW DNA from four filters using a gentle enzymatic lysis based on reference (63) followed by a bead-based cleanup (64). The extracted DNA was further size-selected using a Circulomics Short Read Eliminator XS kit (deplete DNA fragments <10 kbp), yielding a total of 4.1 µg of size-selected DNA. The quality of the DNA was checked on a Nanodrop to ensure that the $OD_{600}$ 260:230 and 260:280 ratios were >1.8,

and the DNA length distribution was checked using a Femto Pulse (Agilent Technologies; Fig. S1).

## Nanopore library preparation and sequencing

Approximately 1 µg of size-selected DNA was used as input to each of four library preps. We generally followed published Oxford Nanopore Technology protocols but extended incubation times. First, we did the DNA repair and end prep using New England Biolabs NEBNext FFPE DNA Repair Mix and NEBNext Ultra II End Repair/dA-Tailing Module. The reactions were 48 µL of size-selected DNA (DNA CS was omitted), 3.5 µL of NEBNext FFPE DNA Repair Buffer, 2 µL of NEBNext FFPE DNA Repair Mix, 3.5 µL of Ultra II End-prep reaction buffer, and 3 µL Ultra II End-prep enzyme mix. The reaction was carried out in a thermocycler for 25 minutes at 20℃ followed by 5 minutes at 65℃. The reaction was cleaned up by using Ampure XP beads at a 1:1 ratio (60 µL) and with two 70% ethanol washes. To elute the DNA, the beads were incubated in water for 10 minutes at room temperature.

Next, the library was prepared using the SQK-LSK110 kit for R9.4 flow cells or SQK-LSK112 kit for R10.4 flow cells. The adapter ligation reaction was extended to 30 minutes. The reaction was cleaned up by using Ampure XP beads at a 2:3 ratio (40 µL) and washed twice with the supplied long-fragment buffer to enrich for DNA fragments longer than 3 kbp. DNA was eluted from beads with 16 µL of elution buffer and incubated at 37℃ for 10 minutes. The library was run on a Femto Pulse to confirm library quality (Fig. S2). ONT long-read sequencing was done using two R9.4.1 MinION flow cells and two R10.4 PromethION flow cells. All runs were set to 72 hours and allowed to go to completion.

## Basecalling, polishing, and assembly of Nanopore data

Basecalling was performed using the Guppy version 6.2.1+6588110a6. The MinION runs used the r9.4.1_450bps_sup model and the PromethION ones used the r10.4_450bps_hac model. We obtained a total of 149,898,872,235 bases (150 Gbp) that passed default Guppy quality control. Read N50/N90 was 11,935/3,570 bases.

The long-read assembly was done using Flye version 2.9.1-b1780 (38). Other than specifying metagenomic mode and –nano-hq for the error rate, all parameters were left at their default settings. Flye selected a minimum overlap of 4,000 bases. The total length of the assembly is 2,948,820,500 bases (2.95 Gbp). There were 107,977 contigs; the N50 was 62,203 bases; the longest contig was 3,483,052 bases (3.5 Mbp); and the average coverage was 36×.

The long-read assembly was error corrected using Medaka version 1.7.1 with model r104_e81_hac_g5015 (https://github.com/nanoporetech/medaka). We selected the model per ONT recommendations based on the available models.

## Illumina metagenomics sequencing and assembly

In addition to the long-read sequencing, we also generated 21 Gbp of 2 × 150 bp reads using Illumina sequencing using DNA from the same extraction. DNA was slightly sheared by pipetting up and down 10 times and was sent to Novogene Corporation Inc. (California, USA) for library prep and 2 × 150 bp sequencing on a NovaSeq6000 (Illumina, USA). The run resulted in 21 Gbp of data with 140,278,770 raw reads, 92.2% reads with quality scores of >Q30. The Illumina reads were trimmed and filtered using BBtools version 38.96 as described in reference 21.

The Illumina reads were assembled using SPAdes version 3.15.5 (39). The total length of the assembly is 948,090,524 bases (0.95 Gbp). There were 813,394 contigs; the N50 was 1,270 bases; the longest contig was 194,182 bases; and the average coverage was 22.15×. These numbers were calculated using Quast (65) with default parameters.

## Protein and RNA gene annotation

Gene calling was done using Prodigal version 2.6.3 (66) for both Nanopore and Illumina assemblies. For both data sets, Prodigal was set to not call genes across edges of contigs, to force a full motif scan, and to use metagenomic mode. For the Nanopore assembly, Prodigal found 4,153,507 genes. For the Illumina assembly, Prodigal found 5,492,197 genes. Both Nanopore and Illumina assemblies were annotated against the full Rfam database using the Infernal package (53) with default parameters. Taxonomy of SSUs was done using the SILVA website (https://www.arb-silva.de [48, 67]) and manual comparison with BLAST searches.

## Taxonomy assignment of contigs

We used MMSeqs2 (68) with default parameters to assign taxonomy to the Nanopore contigs. Out of the total of 107,977 contigs, 95,282 were assigned a taxonomy and 22,761 were assigned down to the species level. We also used Kaiju (69) with default parameters and the db_nr_euk database.

## Taxonomy assignment and completeness check for bins

We used GTDB-Tk version 2 (47) for assigning taxonomy to both complete genomes and bins. For completeness determination, we used CheckM2 (55). In both cases, we ran with default parameters.

## Plasmid and virus classification and analysis

We used geNomad (70) with default parameters to classify contigs as plasmids and viruses. Prophages are noted in geNomad output, and these were excluded from the viral contig counts. We determined viral populations as sequences that had ≥95% ANI as defined by reference (59). We used dedupe.sh from BBtools (BBMap, Bushnell B.; sourceforge.net/projects/bbmap/) using the parameter minidentity = 95. Sequences that were reported as duplicates and contained sequences were removed from the total number of sequences to get the number of viral populations.

## Tree building and visualization

All trees were built using IQTree (71) with the model set to GTR + I + G and the number of fast bootstraps set to 3,000. The trees we were dealing with were relatively small, and we did not see a need for model finding. iTOL (72) was used for tree visualization.

### ACKNOWLEDGMENTS

We thank USGS for collecting samples for us, especially Erica Nejad and the crew of USGS R/V David H. Peterson. We cannot overemphasize the value of the support we have received for this project from USGS. We also thank the Joint Genome Institute for the support that we have received for the PromethION sequencing and library quality analysis. Specifically, we thank Hope Hundley, Rob Egan, Len Pennacchio, and Ronan O'Malley.

This work was supported by the Laboratory Directed Research and Development Program of Lawrence Berkeley National Laboratory under the U.S. Department of Energy (DOE) (Contract No. DE-AC02-05CH11231). Part of this work (proposal: https://doi.org/10.46936/10.25585/60008607) was conducted at the U.S. DOE Joint Genome Institute (https://ror.org/04xm1d337), a DOE Office of Science User Facility supported by the Office of Science of the U.S. Department of Energy operated under Contract No. DE-AC02-05CH11231.

L.M.L. and T.N.N. both conceived the experimental design, processed the samples, analyzed the data, and wrote the manuscript.

## AUTHOR AFFILIATION

[1]Environmental Genomics and Systems Biology Division, Lawrence Berkeley National Laboratory, Berkeley, California, USA

## PRESENT ADDRESS

Torben N. Nielsen, Jorgmundir, Seattle, Washington DC, USA

## AUTHOR ORCIDs

Lauren M. Lui  http://orcid.org/0000-0001-8720-5268
Torben N. Nielsen  http://orcid.org/0000-0002-0987-7189

## FUNDING

| Funder | Grant(s) | Author(s) |
|---|---|---|
| U.S. Department of Energy (DOE) | DE-AC02-05CH11231 | Lauren M. Lui |
| | | Torben N. Nielsen |

## AUTHOR CONTRIBUTIONS

Lauren M. Lui, Conceptualization, Data curation, Formal analysis, Funding acquisition, Investigation, Methodology, Project administration, Visualization, Writing – original draft, Writing – review and editing | Torben N. Nielsen, Conceptualization, Data curation, Formal analysis, Funding acquisition, Investigation, Methodology, Project administration, Software, Writing – original draft, Writing – review and editing

## DATA AVAILABILITY

Nanopore sequencing data are deposited under National Center for Biotechnology Information BioProject PRJNA998863. The refined genome bins from the nanopore metagenome assemblies are available on Zenodo (https://zenodo.org/records/13227923). The Illumina sequencing data and metagenome assemblies are also available on Zenodo (https://zenodo.org/records/13228283).

## ADDITIONAL FILES

The following material is available online.

### Supplemental Material

**Supplemental material (mSystems00242-24-s0001.pdf).** Figures S1-S10 and supplemental methods.
**Table S1 (mSystems00242-24-s0002.xlsx).** SSU taxonomic classification for Bacteria, Archaea, Eukarya, mitochondria, and chloroplasts.
**Table S2 (mSystems00242-24-s0003.xlsx).** Bacterial and archaeal SSU classification by MMSeqs2 using GTDB as the database.
**Table S3 (mSystems00242-24-s0004.xlsx).** geNomad classification results for contigs classified as plasmids (Nanopore assembly).
**Table S4 (mSystems00242-24-s0005.xlsx).** geNomad classification results for contigs classified as plasmids (Illumina assembly).
**Table S5 (mSystems00242-24-s0006.xlsx).** Kaiju classification of the eukaryotic contigs.
**Table S6 (mSystems00242-24-s0007.xlsx).** SSU taxonomic classification for Illumina assembly.
**Table S7 (mSystems00242-24-s0008.xlsx).** Bin quality and taxonomic classification.

Open Peer Review

**PEER REVIEW HISTORY (review-history.pdf).** An accounting of the reviewer comments and feedback.

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
