## [Reviewer comments · mSystems]

Decomposing a San Francisco estuary microbiome using long read metagenomics reveals species- and strain-level dominance from picoeukaryotes to viruses

Lauren Lui and Torben Nielsen

Corresponding Author(s): Lauren Lui, E O Lawrence Berkeley National Laboratory

Review Timeline:

Submission Date:	February 22, 2024
Editorial Decision:	April 22, 2024
Revision Received:	June 21, 2024
Accepted:	July 11, 2024

Editor: Xiao-Hua Zhang

Reviewer(s): Disclosure of reviewer identity is with reference to reviewer comments included in decision letter(s). The following individuals involved in review of your submission have agreed to reveal their identity: Alexander N Ignatov (Reviewer #1)

Transaction Report:

DOI: <https://doi.org/10.1128/msystems.00242-24>

Re: mSystems00242-24 (Decomposing a San Francisco estuary microbiome using long read metagenomics reveals species- and strain-level dominance from picoeukaryotes to viruses)

Dear Dr. Lauren M Lui:

Revision Guidelines

Sincerely,
Xiao-Hua Zhang
Editor
mSystems

Reviewer #1 (Comments for the Author):

Ocean and estuarine microbiomes cause high interest because of their fundamental roles in global element cycling. San Francisco Estuary, the largest estuary on the west coast of the United States has high nutrient loadings that are higher than other estuaries. This along with increased algal toxins and primary production in recent years supports the hypothesis that the San Francisco Estuary is the best model to investigate the health of the Ocean ecosystems.

The manuscript titled "Decomposing a San Francisco estuary microbiome using long read metagenomics reveals species- and strain-level dominance from picoeukaryotes to viruses" is devoted to the study of metagenome to obtain a more accurate estimate of microbial diversity. To achieve this, a new bead-based DNA extraction method was developed, a novel bin refinement method designed, and 150 Gbases of Nanopore sequencing were obtained. An estimated ~500 bacteria and archaeal species in the sample, and 68 high-quality bins (>90% complete, <5% contamination, {less than or equal to}5 contigs, contig length >100 Kbases, and all ribosomal and tRNA genes) were obtained. Many contigs of picoeukaryotes, environmental DNA of larger eukaryotes such as mammals, complete mitochondrial and chloroplast genomes were obtained. ~40,000 viral populations were detected too. Ocean and estuarine microbiomes play critical roles in global element cycling. New sampling and bioinformatics methods to attempt decomposing an estuarine microbiome into its constituent genomes. The results suggest there are only a few strains that comprise most of the species abundances from viruses to picoeukaryotes. Characterization of a metagenome of this diversity requires 1Tbase of long read sequencing. The manuscript is well-written, the presented data support conclusions, and this work can be very important for further development of metagenome and ecosystems analysis.

Reviewer #2 (Comments for the Author):

In this article, Lui and Nielsen describe a detailed and thorough interrogation of the microbial composition of a single water sample from the San Francisco Estuary using nanopore sequencing. Their observations are interesting and the assembly methods are rigorous, but the objectives and takeaways from this n of 1 study are not entirely clear. It is also not clear if it is intended to be presented as a methods study demonstrating improvement in genomic methodologies, or as a results paper presenting new observations about the estuary community.

My major recommendation would be to strengthen the validation of the methods demonstrated in the paper by comparing the findings more thoroughly with what would be found by using Illumina sequencing alone and/or with more standard nanopore methods (e.g. fully automated binning, a single classification tool). To claim that these results are "advances in the use of long reads to obtain genomes from metagenomes", there should be a more detailed comparison with previous methods to put the authors' findings into context. This could include comparing overall properties of the assembly as well as more detailed features. For example, plasmids are thought to be retained better by Illumina sequencing, since they can be lost from a nanopore library prep due to circularity and/or size selection. It would be very interesting to compare plasmids detected in your Illumina data with those found in the nanopore assembly. Similarly, there is a nice comparison with the Illumina taxonomic assignments in lines 269-275, but the details of this comparison (the higher-level Illumina classifications, or which classifications are gained and lost with each method) don't appear to be actually included in the results anywhere.

Other major comments:

-The introduction mentions the importance of high nitrogen and phosphorus loadings in this environment and the possibility of the estuary being at a "tipping point". It would be helpful if the analysis and/or discussion could address whether/how the improved assembly helps address these questions (perhaps gaining deeper insights into the set of organisms with particular metabolic capabilities).

-There are occasional value judgment statements presented without concrete support or details. For example:
-lines 152-154 "Sorting through the different classification methods also points out the hazards of relying on one method" - what are the hazards?
-lines 290-294: "We were selective about choosing binning software...eventually we settled on GraphMB" How/why was GraphMB chosen?

-Are the authors planning to make code available for their binning workflow? (since that is one of the main advances in this paper)

-I recommend the authors reframe their conclusion to focus more on the major takeaways of the study and how they might be useful for other researchers, instead of previewing next steps/unpublished data by the authors.

Minor comments:

-Define SSU when it is first used (line 133?). I would recommend referring to these as "SSU genes" for clarity but not necessary.
-lines 326-328: It would be helpful to explain a little more specifically why SAR11 genomes are difficult to assemble
-lines 342-346: Did the authors distinguish integrated prophages when classifying contigs as viral? (other than based on the presence of SSU genes)
-The "Results and Discussion" section includes short descriptions of the sampling, sequencing, and taxonomic classification, but not of the assembly and/or polishing. It would help with clarity to add a sentence or two to fill in this gap.
-There is some assembly jargon that should be defined for the more general mSystems reader, e.g.:
line 170: "taxonomic collapse"

Line 304: "chaos" bins

-I recommend replacing "manpower/man hours" in line 394 with a gender neutral term such as "labor", "person hours".

Reviewer #1 (Comments for the Author):

Ocean and estuarine microbiomes cause high interest because of their fundamental roles in global element cycling. San Francisco Estuary, the largest estuary on the west coast of the United States has high nutrient loadings that are higher than other estuaries. This along with increased algal toxins and primary production in recent years supports the hypothesis that the San Francisco Estuary is the best model to investigate the health of the Ocean ecosystems. The manuscript titled "Decomposing a San Francisco estuary microbiome using long read metagenomics reveals species- and strain-level dominance from picoeukaryotes to viruses" is devoted to the study of metagenome to obtain a more accurate estimate of microbial diversity. To achieve this, a new bead-based DNA extraction method was developed, a novel bin refinement method designed, and 150 Gbases of Nanopore sequencing were obtained. An estimated ~500 bacteria and archaeal species in the sample, and 68 high-quality bins (>90% complete, <5% contamination, {less than or equal to}5 contigs, contig length >100 Kbases, and all ribosomal and tRNA genes) were obtained.

Many contigs of picoeukaryotes, environmental DNA of larger eukaryotes such as mammals, complete mitochondrial and chloroplast genomes were obtained. ~40,000 viral populations were detected too. Ocean and estuarine microbiomes play critical roles in global element cycling. New sampling and bioinformatics methods to attempt decomposing an estuarine microbiome into its constituent genomes.

The results suggest there are only a few strains that comprise most of the species abundances from viruses to picoeukaryotes. Characterization of a metagenome of this diversity requires 1Tbase of long read sequencing.

The manuscript is well-written, the presented data support conclusions, and this work can be very important for further development of metagenome and ecosystems analysis.

Response:

We thank the reviewer for their comments.

Reviewer #2 (Comments for the Author):

In this article, Lui and Nielsen describe a detailed and thorough interrogation of the microbial composition of a single water sample from the San Francisco Estuary using nanopore sequencing. Their observations are interesting and the assembly methods are rigorous, but the objectives and takeaways from this n of 1 study are not entirely clear. It is also not clear if it is intended to be presented as a methods study demonstrating improvement in genomic methodologies, or as a results paper presenting new observations about the estuary community.

Comment #1:

My major recommendation would be to strengthen the validation of the methods demonstrated in the paper by comparing the findings more thoroughly with what would be found by using

Illumina sequencing alone and/or with more standard nanopore methods (e.g. fully automated binning, a single classification tool). To claim that these results are "advances in the use of long reads to obtain genomes from metagenomes", there should be a more detailed comparison with previous methods to put the authors' findings into context. This could include comparing overall properties of the assembly as well as more detailed features. For example, plasmids are thought to be retained better by Illumina sequencing, since they can be lost from a nanopore library prep due to circularity and/or size selection. It would be very interesting to compare plasmids detected in your Illumina data with those found in the nanopore assembly.

Similarly, there is a nice comparison with the Illumina taxonomic assignments in lines 269-275, but the details of this comparison (the higher-level Illumina classifications, or which classifications are gained and lost with each method) don't appear to be actually included in the results anywhere.

Response:

We appreciate the reviewer's comments in regards to putting the Nanopore assemblies in context with Illumina assemblies. We had previously avoided comparison of the Illumina and Nanopore assemblies because of the vast difference in sequencing effort (150Gbp vs 21Gbp), as this will affect the quality of the assembly and makes comparisons difficult. In general, the shorter average length of the contigs in the Illumina assembly also make it more difficult to classify the contigs as plasmids or to assign taxonomy. We still think that these points hold true, but we also think that the suggestion of adding in additional comparison to the Illumina assembly provides useful analysis and discussion and have added in additional text on plasmids (lines 211-226) and taxonomic assignments of SSUs (lines 323-326).

Comment #2:

-The introduction mentions the importance of high nitrogen and phosphorus loadings in this environment and the possibility of the estuary being at a "tipping point". It would be helpful if the analysis and/or discussion could address whether/how the improved assembly helps address these questions (perhaps gaining deeper insights into the set of organisms with particular metabolic capabilities).

We have added additional text to the introduction to clarify this point and explain the potential impact of improved assembly in relation to biogeochemical cycling studies (lines 69-78).

-There are occasional value judgment statements presented without concrete support or details.

We thank the reviewer for pointing out the statements below as the manuscript benefits from the clarifying text. We have included explanations below and the corresponding edited lines in the revised manuscript.

For example:

-lines 152-154 "Sorting through the different classification methods also points out the hazards of relying on one method" - what are the hazards?

In more than one case, the database used had a strong influence on the results. For example, if only MMseqs2 with the GTDB database was used for classification, then ~85k contigs were classified as bacterial. This is in contrast with our final evaluation, where we estimate that only ~41K contigs were bacterial after using all of the different classification methods. We have added more description in the main text to clarify this point (lines 161-168, line 180).

-lines 290-294: "We were selective about choosing binning software...eventually we settled on GraphMB" How/why was GraphMB chosen?

Older metagenomics bidders do not take full advantage of assembly graphs which are available from modern assemblers. They are also not optimized for long read assemblies. GraphMB employs graph neural networks that take full advantage of the assembly graph and the long reads. We have added text to address this point (lines 345-346).

-Are the authors planning to make code available for their binning workflow? (since that is one of the main advances in this paper)

We would like to make all of the code we use available. However, it is not in the form of a finished script that can run automatically. Rather it is a collection of little pieces that are run iteratively and require human intervention between iterations. Longer term, we are working on ways of building automated work flows to help with this.

-I recommend the authors reframe their conclusion to focus more on the major takeaways of the study and how they might be useful for other researchers, instead of previewing next steps/unpublished data by the authors.

We have added more text in regards to major takeaways on lines 363-368.

Minor comments:

-Define SSU when it is first used (line 133?). I would recommend referring to these as "SSU genes" for clarity but not necessary.

This edit has been added to the manuscript (line 151-152).

-lines 326-328: It would be helpful to explain a little more specifically why SAR11 genomes are difficult to assemble

We have added additional text in regards to this point (392-395).

-lines 342-346: Did the authors distinguish integrated prophages when classifying contigs as viral? (other than based on the presence of SSU genes)

Since we used geNomad to predict if contigs were viral, it classifies whether a contig contains a prophage or not. These were excluded from the viral contig counts. We have added clarifying text (lines 572-573).

-The "Results and Discussion" section includes short descriptions of the sampling, sequencing, and taxonomic classification, but not of the assembly and/or polishing. It would help with clarity to add a sentence or two to fill in this gap.

Additional text has been added to describe the assembly and polishing steps (lines 128-132).

-There is some assembly jargon that should be defined for the more general mSystems reader, e.g.:

line 170: "taxonomic collapse"

Line 304: "chaos" bins

We thank the reviewer for pointing this out and have added clarifying text for the above jargon (lines 197-199, 357-365).

-I recommend replacing "manpower/man hours" in line 394 with a gender neutral term such as "labor", "person hours".

We appreciate this comment about gender neutral terms and have changed it to "person" hours (line 468).

Re: mSystems00242-24R1 (Decomposing a San Francisco estuary microbiome using long read metagenomics reveals species- and strain-level dominance from picoeukaryotes to viruses)

Dear Dr. Lauren M Lui:

I am happy with the current version.

Your manuscript has been accepted, and I am forwarding it to the ASM production staff for publication. Your paper will first be checked to make sure all elements meet the technical requirements. ASM staff will contact you if anything needs to be revised before copyediting and production can begin. Otherwise, you will be notified when your proofs are ready to be viewed.

Sincerely,
Xiao-Hua Zhang
Editor
mSystems

Reviewer #1 (Comments for the Author):

The manuscript "Decomposing a San Francisco estuary microbiome using long read metagenomics reveals species- and strain-level dominance from picoeukaryotes to viruses" has received a high score in the first round of review. There is no doubt that it would be even better after some revisions. There is no doubt that it is even better after some revision.

Reviewer #2 (Comments for the Author):

I thank the authors for their thoughtful responses - the paper is much improved. The comparison of plasmid annotations between the two assemblies is very interesting. Two comments:

-Line 461 still discusses "manhours" although the line above it was changed to "person hours"

-I would still strongly recommend that the authors make their binning code publicly available for reference (consistent with mSystems policy), even if not in the form of a fully polished software tool.